# The Successful Treatment of a Patient with Ehlers–Danlos Syndrome (EDS) After an Extensive Burn Injury: A Case Report

**DOI:** 10.3390/medicina61040554

**Published:** 2025-03-21

**Authors:** Karolina Ziółkowska, Anna Słaboń, Justyna Glik, Mariusz Maj, Magdalena Olszak, Karolina Mikuś-Zagórska, Przemysław Strzelec, Katarzyna Czerny, Ryszard Maciejowski, Marcin Gierek, Wojciech Łabuś

**Affiliations:** 1Stanislaw Sakiel Burn Treatment Center in Siemianowice Śląskie, Jana Pawła II Street 2, 41-100 Siemianowice Śląskie, Poland; 2Doctoral School of the Medical University of Silesia in Katowice, Faculty of Health Sciences in Bytom, Medical Uniwersity of Silesia in Katowice, 41-902 Bytom, Poland; 3Faculty of Health Sciences in Katowice, Medical University of Silesia in Katowice, 40-752 Katowice, Poland

**Keywords:** Ehlers–Danlos Syndrome (EDS), burn wounds, acellular dermal matrix (ADM), in vitro cultured skin cell application, MEEK technique, laser speckle contrast analysis, surgical treatment

## Abstract

*Introduction*: Ehlers–Danlos Syndromes (EDSs) are a heterogeneous group of monogenic connective tissue disorders (e.g., joint hypermobility and dislocation, skin hyperelasticity and fragility, chronic pain, delayed wound healing process,, etc.). The primary objective of this study was to present a specialized therapeutic wound management process for a burn-injured female patient diagnosed with EDS. *Case Presentation*: A 34-year-old female patient presented with extensive thermal burns (biofireplace explosion). The patient had a family history of diagnosed EDS. Additionally, the patient was in a poor mental condition and, since 2020, had been undergoing pharmacotherapy with antidepressant and anti-anxiety medication. This might be the first such clinical observation in the world, but a correlation has been observed between psychiatric medication use and EDS wound healing impairment. During the hospitalization process, the patient underwent a series of surgeries aimed at the fastest and most effective closure of wounds. The patient, after 182 days of hospitalization in our facility, was discharged home. *Materials and Methods*: During the patient’s hospital stay, the patient underwent multiple procedures involving debridement of necrotic tissues. Additionally, allogeneic acellular dermal matrix (ADM) grafting was performed on the wounds, and a procedure was conducted in which skin was grafted using the MEEK technique. The in vitro cultured skin cells, as the advanced therapy medicinal products (ATMPs), were used. During the patient’s stay in the hospital, images were taken using low-energy laser speckle contrast analysis (LASCA) to asses microperfusion or lack thereof. The measurements were taken at intervals of several days. *Conclusions*: The treatment of burn wounds in patients with EDS requires a long hospitalization period. It also may require a multi-stage approach utilizing innovative preparations (e.g., ADMs and ATMPs). The assessment of wound healing progress can be performed using advanced equipment, such as laser speckle contrast analysis (LASCA).

## 1. Introduction

Ehlers–Danlos syndrome (EDS) encompasses a heterogeneous group of inherited connective tissue diseases that are estimated to affect about 1 in 5000 people worldwide. Specific genetic defects in the biosynthesis or structure of collagen can be considered as the main cause of EDS. According to the 2017 International Classification of EDS, there are 13 clinical subtypes of EDS [1]. Additionally, there are data on the classification criteria for individual EDS subtypes (all data are available in the Appendix A) [1]. However, crucially, a clinical assessment of symptoms is sufficient for the clinical diagnosis of EDS. In this manner, the phenotypic characteristics of this disorder are tissue fragility and joint hypermobility and hyperstretching of the skin, with abnormal scar formation and other features such as aortic dissection, fragility of blood vessels, heart valves, rupture of specific organs, and often chronic pain. Other characteristic features of EDS include thin skin, atrophic scarring, and easy bruising, with inguinal hernia, chest deformity (especially funnel deformity), joint dislocations, and foot deformities (e.g., flat foot, plano foot valgus, and big-toe Hallux valgus). However, in order to determine the EDS subtype in detail, it may be necessary to perform detailed genetic tests [1,2,3,4,5,6,7].

This paper presents the case report of a 34-year-old female patient with chronic wounds which resulted from a thermal burn involving 60% of the total body surface area (TBSA), classified as II/III degree burns. The patient had a confirmed prior diagnosis of EDS and a positive family history of the condition.

A standard approach for the treatment of the extensive and deep-to-full-thickness burns involves early excision and transplantation of autologous split-thickness skin grafts (STSGs). In cases of extensive and deep burns and a lack of suitable donor sites, there might be a need for grafting using a widely meshed STSG. However, due to the uneven healing of the interstices and the adhered graft, this technique often results in wound patterning and delayed healing. A prolonged healing time can elevate the risk of infection and increase the need for intensive wound care, while wound patterning may have significant psychological and esthetic implications for patients. In order to overcome these limitations, an in vitro cultured autologous skin cell suspension can be applied as an “over spray” to a meshed autograft. One of the proposed methods for applying in vitro cultured fibroblasts and keratinocytes is the administration of a suspension of these cells in platelet-rich plasma (PRP). Such a procedure could accelerate wound healing and reduce the extent of patterning in healed burn wounds. It should also be remarked that in vitro cultured skin cells (keratinocytes and dermal fibroblasts) are classified as advanced therapy medicinal products (ATMPs) [8,9].

Another alternative prospect for managing extensive and deep burns is the Meek technique, which has significant potential as a valuable tool in burn wound treatment. This approach offers an effective strategy for enhancing patient outcomes in complex burn injuries [9].

In this context, it is important to emphasize that performing an autologous STSG, whether using the meshing method or the Meek technique, aims to maximize the surface area of the graft as much as possible. However, this comes at the cost of graft integrity loss, which may consequently lead to cosmetically and functionally unacceptable scarring. Nevertheless, in the desperate fight for the life of a patient with extensive and deep burns, this approach may be the only viable option [8,9,10,11].

In this respect, a promising procedure that could significantly improve the characteristics of the wound after the deep excisions could be the transplantation of a acellular dermal matrix (ADM) as the co-graft for the autologous STSG [10,11].

The healing process of burn wounds and donor sites necessitates the use of appropriate, validated methods to objectively evaluate local wound conditions. Several well-established approaches exist for documenting and monitoring wound healing progress. To determine the extent of a burn and subsequently track the effects of applied therapies, techniques such as laser Doppler devices or laser speckle contrast analysis (LASCA) can be utilized, providing an objective assessment of the wound’s blood supply [12,13,14].

In this context, it can be observed that the therapeutic management of extensive and deep burns presents a significant clinical challenge [8,9,10,11,14,15,16,17]. In burn patients with a primary diagnosis of EDS, the wound healing process may be further substantially impaired. Additionally, literature reports have described cases in which specific pharmacological agents may exert a negative impact on wound healing in patients with EDS [18].

The objective of this study was to present a specialized therapeutic wound management process for an extensive and deep burn-injured female patient diagnosed with EDS.

## 2. Materials and Methods

The medical procedures used in this study were performed in accordance with the ethical standards of the Declaration of Helsinki (1964, last amended in 2008).

All the activities performed were recognized and formally approved forms of treatment. In addition, all the activities performed were aimed at improving the patient’s health, and in certain cases, constituted a desperate attempt to save her health and life. According to Polish law, retrospective observational studies, including case reports, do not require approval from a bioethics committee.

The patient gave informed consent to hospitalization and to the publication of medical data and images.

### 2.1. Case Presentation

A 34-year-old female patient was transported to the Stanisław Sakiel Burns Treatment Center in Siemianowice Śląskie, Poland, in June 2022 due to multiple burn wounds on the left upper limb, left thigh, right thigh, shin, abdominal wall, and buttocks. Additionally the patient suffered from a wound on the abdomen following numerous laparotomies.

The patient underwent her first hospitalization due to a burn injury in June 2021 at the Center for the Treatment of Severe Burns in Gryfice, Poland, where split-thickness skin grafts were performed. In the following months, the patient was hospitalized in several other medical facilities due to complications during the course of treatment.

Additionally, the patient exhibited symptoms of depression and suffered from hypothyroidism in the course of Hashimoto’s disease.

The chronic wounds resulted from a thermal burn suffered in June 2021, involving 60% of the body surface area, classified as II/III degree burns. During the hospitalization, she was treated in the general surgery department. She remained hospitalized from June to September 2022, when she had to be transferred to the intensive care unit due to septic shock, consciousness disorders, and hypodynamic respiratory failure.

Throughout the patient’s stay in the General Surgery Department, 20 surgical procedures were performed, spaced approximately one week apart. During these procedures, surgeons performed wound debridement. During the first procedure, an ADM was applied to the granulating wound on the left lower limb, which was dressed with paraffin gauze (Jelonet, Smith & Nephew, London, UK) and antiseptic (SutriSept^®^, Warsaw, Poland). Iodine dressings (Braunol, Melsungen, Germany) were applied to the remaining wounds.

Due to unstable mental condition and symptoms of depression, the patient received psychotropic medications (Trittico, Pramolan, Pregabalin).

During routine wound care while hospitalized in the surgical ward, the patient reported severe pain, necessitating the implementation of ad hoc analgesic therapy.

On the twelfth day after admission to the General Surgery Department, a decision was made to administer a vaccine against *Pseudomonas aeruginosa* (Pseudovac, IBSS BIOMED, Krakow, Poland).

A decision was made to provide an in vitro skin cell culture (autologous keratinocytes and dermal fibroblasts). During the second surgical procedure, a small skin graft (approx. 2 cm × 2 cm) was obtained using a battery dermatome (Aesculan III, B.Braun). A skin fragment was donated from the right foot, where the skin was intact. The fragment of skin was transferred to the tissue bank located at the Stanisław Sakiel Burns Treatment Center. The in vitro cell culture was carried out in accordance with the standard operating procedures of the tissue bank under Good Manufacturing Practice (GMP) conditions. After the skin procurement, a polyurethane foam dressing with silver ions (Allevyn, Smith & Nephew) was applied to the donor site. During that procedure, the allogeneic skin grafts were applied to the cleaned surfaces of the left arm and the right shin. The wounds were dressed as mentioned before (with paraffin gauze and iodine antiseptic).

Subsequent procedures involved wound debridement, allogeneic skin grafts, and collagen injection around the wound edges (MD-Matrix, Milan, Italy). Paraffin-impregnated gauze dressings (Jelonet, Smith & Nephew), as the standard of care, and two alternating antiseptics (SutriSept^®^, Verco; Prontosan, B.BRAUN) were used for wound management.

Due to the deterioration of the patient’s condition, she was transferred to the intensive care unit (ICU). Parenteral nutrition was initiated. All psychotropic medications previously taken by the patient (Trittico, Pramolan, Pregabalin) were discontinued. During the stay in the ICU, two surgical wound debridement procedures were performed, with a one-week interval between procedures. During the second procedure, ADM was applied to the debrided wounds. After 15 days, the patient was transferred back to the general surgery department, where surgical wound debridement, collagen injection around the wound edges (MD-Matrix, GUNA), the closure of wounds using autologous grafts, and dressing of wounds with standard-of-care dressings (Jelonet, Smith & Nephew) and antiseptic (SutriSept^®^, Verco; Prontosan, B.BRAUN; Braunol, B.BRAUN) were continued (Figure 1).

In the fifth month of hospitalization, a procedure was conducted in the operating theater to address the wound on the posterior surface of the left thigh. During this procedure, collagen was injected and autologous skin cells (keratinocytes and dermal fibroblasts) that had been cultured in vitro and suspended in platelet-rich plasma (PRP) solution were applicated. Additionally, the wounds were treated with ADM. During the final operation before discharge, a split-thickness skin graft was harvested and sliced using the MEEK (ang. micrografting method) 1:4 and 1:3 method. PRP was injected into the wounds, and cultured keratinocytes were applied. A split-thickness skin graft prepared using the MEEK method was then applied to the wounds and secured with skin staples (Figure 2). The entire area was covered with paraffin gauze (Jelonet, Smith & Nephew) and treated with antiseptic (Prontosan, B.BRAUN). The patient was discharged home before Christmas 2022.

### 2.2. Imaging Using Laser Speckle Contrast Analysis Technique

LASCA (ang. laser speckle contrast analysis) uses laser light to illuminate the area under examination, where the backscattered light forms an interference pattern consisting of dark (black or dark navy area) and bright regions (shades of blue, green, yellow, orange, or red color. The greater the microperfusion, the warmer the shade of color displayed in the examined area). This pattern, known as a speckle pattern, remains stationary if the illuminated object is static. However, when there is movement within the object, such as moving erythrocytes, the speckle pattern changes.

During the six-month stay of the patient at the Stanisław Sakiel Burn Treatment Center in Siemianowice Śląskie, images were captured using a low-energy laser, highlighting microperfusion in the blood vessels. In our study, we used a laser system (Perimed AB, Järfälla, Sweden) operating in the near-infrared spectrum (NIR, 785 nm) of the electromagnetic spectrum [12]. The images were captured at intervals of several days. Measurements were consistently taken at the same distance from the patient’s body surface (30 cm) and lasted for 20 s each time. The image size was 20.1 cm in width and 19 cm in length.

LASCA is a device used for imaging microperfusion in a specific ROI (Region Of Interest). It is widely used for imaging the degree of burns [13,14] as well as inflammatory conditions in patients with hidradenitis suppurativa [10,11]. It also enables monitoring of the treatment process. The use of this device to monitor the patient with EDS allowed visualization of areas where wound re-opening occurred and of those areas that had already healed (Figure 3.). The microperfusion measurements were performed on the patient throughout the entire period of hospitalization. The measurement results were collected in Figure 4 and Figure 5.

## 3. Results

The patient’s total length of stay in Stanislaw Sakiel Burn Treatment Center in Siemianowice Śląskie was 182 days. During hospitalization, the patient underwent multiple procedures (the patient underwent 20 surgical procedures), including necrotic tissue debridement. Additionally, allogeneic ADM grafting was performed, along with autologous skin transplantation using the MEEK technique. The patient received in vitro cultured skin cells suspended in autologous platelet-rich plasma (PRP). Additionally, the wound healing process was supported by injections of a collagen-containing preparation. Throughout the hospital stay, low-energy laser LASCA imaging was conducted to assess microperfusion or its absence, with measurements taken at several-day intervals.

From the very beginning, the treatment of the patient was not easy. A significant difficulty was the patient’s mental state, resulting from depression issues she had been dealing with since 2020, as well as the very long period of hospitalization. Due to such a prolonged hospital stay and the patient’s low pain threshold, she had to receive increasingly stronger pain medication. Thanks to providing appropriate pain management, it was possible to change dressings, which were present almost all over the body, and to thoroughly cleanse the wounds. Important factors that could influence the wound healing process included the observation of an inhibited healing process while the patient was taking three psychotropic medications simultaneously (with anxiolytic and antidepressant effects), maintaining wound antisepsis, the continuous monitoring of the patient’s condition, combining multiple techniques, and taking an interdisciplinary approach to the treatment process. After 182 days of hospitalization at the Stanislaw Sakiel Burn Treatment Center in Siemianowice Śląskie, the patient was discharged home.

## 4. Discussion

EDS encompasses a broad spectrum of connective tissue disorders and presents with a wide range of possible clinical manifestations (all data are available in the Appendix A) [1,2]. In EDS, a characteristic phenomenon is the presence of a large amount of denatured or fragmented collagen in the extracellular space of the dermis in patients. Thus, the formation of a stable and properly organized collagen structure in the extracellular matrix is severely impaired in patients with EDS [1].

In this context, it is worth noting that EDS may be a factor delaying the wound healing process, including burn wounds. Therefore, it can be hypothesized that patients with EDS, in the event of a burn injury, may require prolonged advanced treatment involving innovative preparations, medicinal products, and medical technologies.

This study presents the treatment outcomes of a patient diagnosed with EDS who sustained a thermal injury. The variety of therapeutic approaches utilized may indicate an attempt to maximize treatment effectiveness in the patient’s case.

Since EDS is a connective tissue disorder, a decision was made to utilize available collagen-based preparations. In this context, the first choice was in-house prepared ADM grafts from the tissue bank. The use of ADM in burn treatment has been a well-established procedure for many years [10,11,19,20,21]. Similarly, the commercially available collagen preparation for injection into the wound area has been widely used [21]. The use of in vitro cultured autologous skin cells [22,23,24] represents a promising therapeutic approach for significantly accelerating burn wound healing. It could be once again remarked that these cells are classified as advanced therapy medicinal products and are utilized in Poland under the hospital exemption framework (ATMP-HE). This treatment has been available for many years and has demonstrated proven efficacy, as documented in previous studies [22,23,24,25,26].

The healing process of burn wounds necessitates the use of appropriate and validated methods for objectively evaluating local wound conditions. Several well-established approaches exist for documenting and monitoring wound healing progress. To determine the extent of the burn and subsequently track the effects of the applied therapy, techniques such as laser Doppler imaging or LASCA can be recommended. These methods provide an objective assessment of wound perfusion and blood supply [14,27].

Based on the clinical observation of the patient’s wound healing progress, it was noted that the greatest influence on the wound healing process in the patient could be exerted by various antidepressant and anxiolytic medications she was taking. There is a lack of literature reports supporting this observation. However, there should be no doubt that the authors of this study, having encountered a rare clinical case with an even rarer drug-related complication, have a duty to report this observation to the scientific community. It is essential to emphasize that the primary limitation of this study is the fact that the presented results and observations pertain to a single clinical case. However, the significant correlation between the use or non-use of psychiatric medications and the wound healing process was so evident that it cannot be overlooked. Moreover, the observed lack of literature supporting these findings may rather be considered evidence that the authors may be the first in the world to highlight this concerning clinical symptom. Additionally, it can be hypothesized that the continuous use of psychiatric medications may have contributed to the recurrence of wounds in the patient after being discharged from the burn center in Gryfice, where her burn injuries had been successfully treated initially. A certain confirmation of this hypothetical assumption may be the fact that after admission to CLO, the patient continued psychotropic therapy. However, after the discontinuation of these medications in the ICU of CLO, a clear and significant improvement in the wound healing process was observed. The lack of confirmation of these observations in the literature does not allow for a definitive conclusion regarding the negative impact of psychiatric medications on the wound healing process in EDS. Nevertheless, there is a study describing the adverse effects of these drugs in this patient population. In the work of Bourji et al., it was sulfonamide-induced leukocytoclastic vasculitis confined to the skin in a patient with EDS was presented, highlighting the importance of a comprehensive diagnostic and exclusion workup [18]. In this particular case, it is also important to mention Lyell’s syndrome (toxic epidermal necrolysis, TEN) and Stevens–Johnson syndrome (SJS). TEN and SJS result from an adverse drug reaction, including, e.g., psychotropic medications, antibiotics, or aminosalicylates; however, the pathophysiology of these conditions is not yet fully understood. TEN is characterized by extensive epidermal and epithelial detachment across large surface areas and is potentially life-threatening [28,29].

The duration of hospitalization and constant monitoring of the patient’s health also played a significant role. The combination of taking psychiatric medications prescribed by a psychiatrist with symptoms of EDS such as excessive skin elasticity and fragility, abnormal scar formation, fragility of blood vessels, and chronic pain [1,2] could have led to the inhibition of the wound healing process. Data obtained from the literature [1,2] indicate that there is a large amount of denatured or fragmented collagen in the extracellular space of fibroblasts in patients with Ehlers–Danlos syndrome. We assume that the formation of a stable extracellular collagen matrix is severely impaired in patients with EDS [1]. Nevertheless, impaired collagen synthesis and enzymatic modification appear to be the main cause of EDS. In the study of Whitaker Iain, Warren Rozen et al., the authors observed the arrangement of collagen fibers in cross-sectional samples viewed under a microscope at magnifications of 50,000 and 55,000. In the images obtained, it can be observed that in the sample from the control patient, collagen fibers are circular and arranged regularly. However, in the sample from the patient with EDS, the fibers were irregularly arranged, thinner, and angled [14].

Ehlers–Danlos syndrome is a condition that affects many structures of the human body [1,2,4,6]. Regardless of the type of EDS, there are certain common features in all cases, such as thin and delicate ‘paper-like skin’, easy bruising, a tendency to form scars, and joint hypermobility [1,2,15,16]. Surgical complications in EDS generally fall into three categories [4]. One of them involves cutaneous complications, including wide scar formation, a poor ability to retain cutaneous sutures, and wound dehiscence. The second regards the vascular complications, including the fragility of large vessels and oozing from smaller blood vessels. And the third one includes gastrointestinal complications involving dehiscence of bowel anastomoses [4].

On the basis of the literature analysis on the skin symptoms of EDS, it was concluded that there is a deficit of literature data on this disorder in the available literature. However, there are data on the classification criteria for individual EDS subtypes (all data are available in the Appendix A) [1]. On the basis of the analysis of the description of clinical symptoms, it is possible to make a diagnosis; however, in order to determine the EDS subtype in detail, it may be necessary to perform detailed genetic tests [1,2,3,4,5]. Nevertheless, based on the obtained results, it can be concluded that burn treatment in patients with Ehlers–Danlos syndrome (EDS) presents a significant therapeutic challenge. Managing such cases may require the use of selected advanced products (e.g., ADM, in vitro cultured skin cells) and innovative methods for assessing wound severity and healing progress (e.g., LASCA).

## 5. Conclusions

The treatment of wounds, particularly in the case of burn wounds, in patients with Ehlers–Danlos syndrome requires a long hospitalization period.

The treatment of burn wounds in patients with Ehlers–Danlos syndrome (EDS) may require a multi-stage approach utilizing innovative preparations (e.g., acellular dermal matrix, ADM) and advanced therapy medicinal products (ATMPs).

The assessment of wound healing progress can be performed using advanced equipment, such as laser speckle contrast analysis (LASCA).

## Figures and Tables

**Figure 1 medicina-61-00554-f001:**
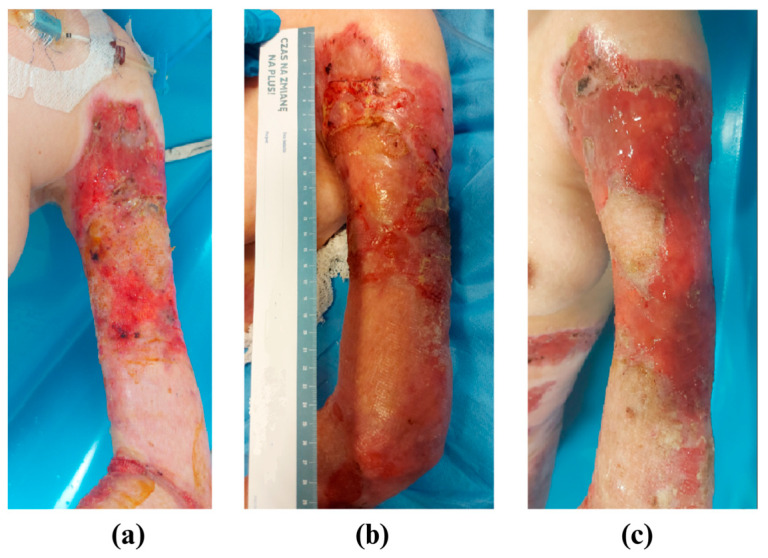
Photos of the left arm. (**a**) The patient during the first stay in the surgical ward; after 2 months of hospital stay, the patient took Trittico, Pramolan, and Pregabalin. (**b**) The patient during her stay in the anaesthesiology and intensive care unit, 3 months after admission. During the stay, no psychotropic drugs were administered. (**c**) The patient after returning to the general surgery ward, 4 months after admission, returned to using Trittico, Pramolan, and Pregabalin.

**Figure 2 medicina-61-00554-f002:**
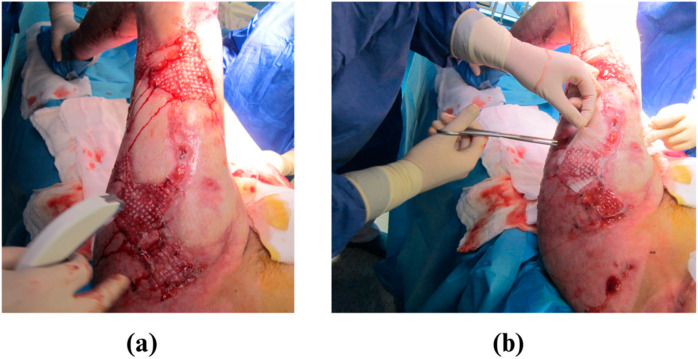
(**a**,**b**) Closure of the wound using a split-thickness skin graft harvested and sliced using the MEEK method in a ratio of 1:4 and 1:3.

**Figure 3 medicina-61-00554-f003:**
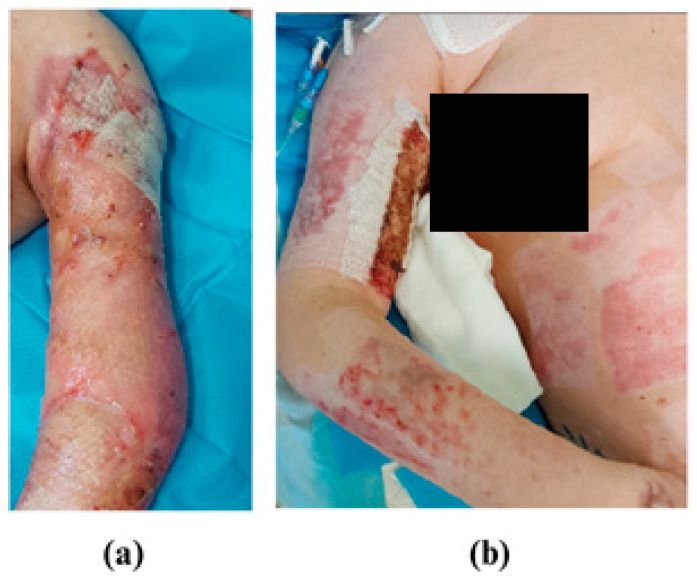
The patient with healed wounds on the day of discharge, (**a**)—left arm, (**b**)—right arm and forearm.

**Figure 4 medicina-61-00554-f004:**
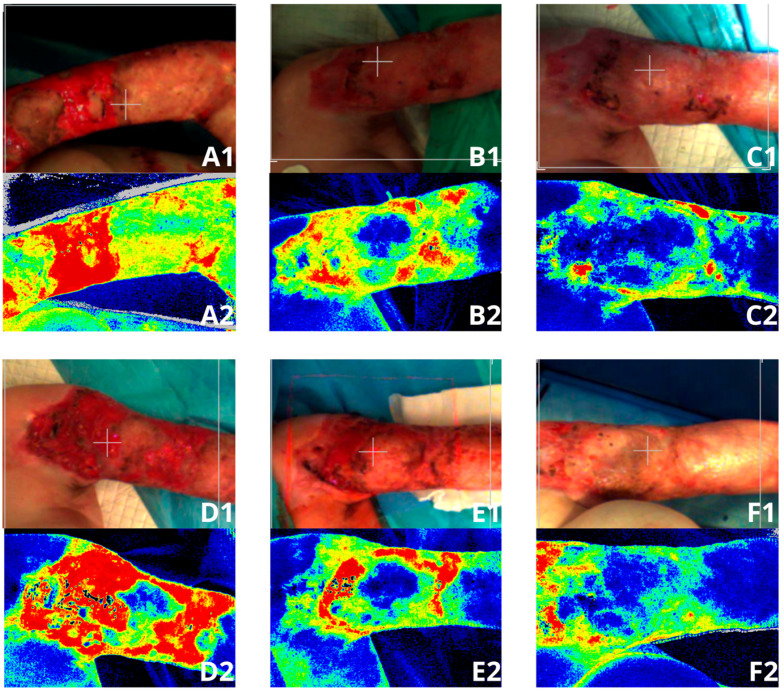
LASCA images of a burn patient suffering from Ehlers–Danlos syndrome (EDS). (**A1**,**A2**)—twenty-seventh day of hospitalization, first measurement of microperfusion, (**B1**,**B2**)—the measurement taken during the patient’s stay in the Department of Anesthesiology and Intensive Care, (**C1**,**C2**)—picture taken at the end of hospitalization in the Department of Anesthesiology and Intensive Care, (**D1**,**D2**)—the measurement taken after the patient’s return from the Department of Anesthesiology and Intensive Care to the Department of General Surgery, where the patient resumed taking medications prescribed by the psychiatrist, (**E1**,**E2**)—the measurement taken on the one hundred and fifty-second day of hospitalization, (**F1**,**F2**)—the measurement taken on the day of the patient’s discharge home.

**Figure 5 medicina-61-00554-f005:**
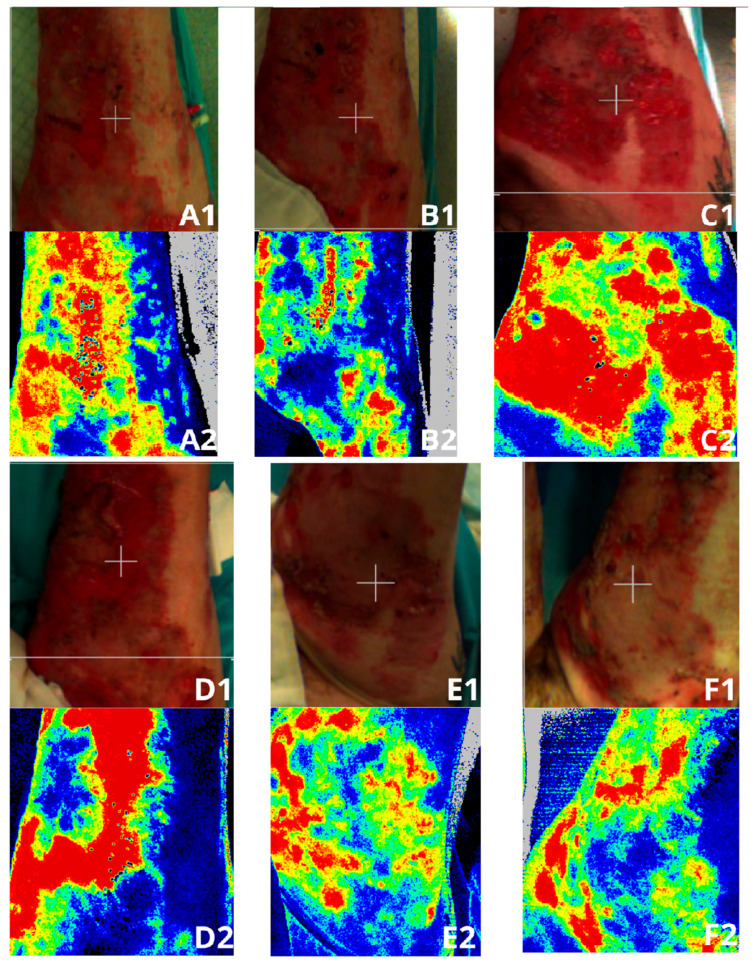
LASCA images of a burn patient suffering from Ehlers–Danlos syndrome (EDS). (**A1**,**A2**)—the measurement taken during the patient’s stay in the Department of Anesthesiology and Intensive Care, (**B1**,**B2**)—a picture taken at the end of hospitalization in the Department of Anesthesiology and Intensive Care, (**C1**,**C2**)—the measurement taken after the patient’s return from the Department of Anesthesiology and Intensive Care to the Department of General Surgery, where the patient resumed taking medications prescribed by the psychiatrist, (**D1**,**D2**)—the measurement taken before the procedure in the operating room, during which the wounds were cleansed, and thereafter where the wound edges were injected with collagen, (**E1**,**E2**)—the measurement taken on the one hundred and seventy-sixth day of hospitalization, (**F1**,**F2**)—the measurement taken on the day of the patient’s discharge home.

## Data Availability

The original contributions presented in this study are included in the article/Appendix A. Further inquiries can be directed to the corresponding authors.

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
