# Peer review of "The Successful Treatment of a Patient with Ehlers–Danlos Syndrome (EDS) After an Extensive Burn Injury: A Case Report"

_medicina, 2025, doi:10.3390/medicina61040554_

Round 1
Reviewer 1 Report
Comments and Suggestions for Authors
The authors present a wound management protocol of a female EDS ( Ehlers Danlos Syndromes) patient with burn wounds. Case presentation: A 34-year-old female patient presented with an extensive thermal burns (biofireplace explosion). Additionally the patient suffered from a chronic, iatrogenic wound and numerous laparotomies with and numerous laparotomies with partial resection of the sigmoid colon and a colostomy. She was in a poor mental state. Since 2020, she has been taking antidepressant and anti-anxiety medication prescribed by a psychiatrist. During the hospitalization process, the patient underwent a series of surgeries aimed at the fastest and most effective closure of wounds. Every interruption in skin continuity poses open gates for infection. The patient, after 182 days of hospitalization in our facility, was discharged home. The patient had a family history of diagnosed EDS.
The manuscript is well written and the figures are clear. The authors justify the choice of therapies very well and the conclusions are described in detail.
Author Response
Comment: "
The authors present a wound management protocol of a female EDS ( Ehlers Danlos Syndromes) patient with burn wounds. Case presentation: A 34-year-old female patient presented with an extensive thermal burns (biofireplace explosion). Additionally the patient suffered from a chronic, iatrogenic wound and numerous laparotomies with and numerous laparotomies with partial resection of the sigmoid colon and a colostomy. She was in a poor mental state. Since 2020, she has been taking antidepressant and anti-anxiety medication prescribed by a psychiatrist. During the hospitalization process, the patient underwent a series of surgeries aimed at the fastest and most effective closure of wounds. Every interruption in skin continuity poses open gates for infection. The patient, after 182 days of hospitalization in our facility, was discharged home. The patient had a family history of diagnosed EDS. The manuscript is well written and the figures are clear. The authors justify the choice of therapies very well and the conclusions are described in detail."
Response: "Dear Reviewer,
We sincerely thank you for the time you have devoted to reviewing our manuscript, as well as for your valuable comments and positive evaluation of our work. We are pleased that you found our study well-written and that the figures and tables presented are clear. We also appreciate your recognition of the justification for the chosen therapies and the detailed description of our conclusions. Your review serves as motivation for us to continue our research and further efforts. Once again, thank you for your time and constructive feedback.
Best regards.
Reviewer 2 Report
Comments and Suggestions for Authors
This is a valuable case report that reports on the treatment progress of a patient with Ehlers-Danlos syndrome (EDS) who sustained burns. However, this paper has the following problems. First, it is not formatted as a case report. Second, the viewpoint of the paper is not specific to EDS, but is more general.
In case reports, it is necessary to avoid general descriptions as much as possible and to proceed with discussion based on the information obtained from the case. It is also necessary to aim for concise and clear descriptions.
Abstract
The following sentences are not necessary in the Abstract or should be expressed concisely.
"joint hypermobility and dislocation, skin hyperelasticity and fragility with abnormal scar formation, and other features such as aortic dissection, blood vessel fragility, organ rupture, chronic pain and often"
"Additionally, the patient suffered from a chronic, iatrogenic wound and numerous laparotomies with partial resection of the sigmoid colon and a colostomy."
This is not directly related to burns or treatment, so the need to include it in the Abstract should be reconsidered.
"She was in a poor mental state. Since 2020, she has been taking antidepressant and anti-anxiety medication prescribed by a psychiatrist."
While the assessment of mental status and the presence or absence of treatment are clinically important, they are not specific to EDS, so their description should be kept to a minimum.
In addition, since "ADM" on line 27 is a first appearance, it should be spelled out in full as "acellular dermal matrix."
Conclusion
The following statements are inappropriate or unnecessary.
"Treating a patient with EDS is not the easiest thing to do."
This is self-evident and not necessary for the conclusion.
"During the patient's hospitalization, many adjustments were made to the treatment, one of the most important being the temporary discontinuation of medications prescribed by a psychiatrist."
This is not specific to EDS, so it is inappropriate for the conclusion.
There have been few reports of ADM or skin grafting in EDS patients, and the usefulness of specific treatments in the treatment of burns in EDS should be described based on the treatment course of this case.
1. Introduction
The purpose of the introduction is to briefly explain the underlying disease that is the background of the case report and to clarify the uniqueness and specificity of this case. However, the introduction of this paper is limited to a detailed explanation of EDS (contents of Tables 1 and 2), and the introduction of LASCA (Laser-Assisted Skin Cell Application) in EDS, as stated in the title of the paper, is missing.
In addition, since this case report does not describe the subtype of EDS or the results of genetic testing, Tables 1 and 2 are considered unnecessary. Instead, the characteristics of LASCA and the circumstances that led to the reporting of this case should be clearly stated as an introduction.
2.1 Case Presentation
The description in 2.1 is redundant and difficult to read. One reason for this is the repeated use of the same words. In particular, the frequent use of "paraffin gauze and antiseptic" should be considered as to whether it is necessary to mention it repeatedly.
In addition, on page 7, line 109, it is stated that "a skin graft was obtained using a dermatome and transferred to the Tissue Bank located at the Burns Treatment Center for cell culture." However, there are few reports of allogeneic skin grafts for EDS patients. As stated in reference 13,
the donor site, total grafted area, and healing time should be detailed.
In addition, Figures 1 to 5 only contain descriptions of the time and location (department) of the images, and there is insufficient explanation of the findings. If there are findings specific to EDS patients, they should be clearly stated. In addition, these figures are not related to psychiatric prescriptions and are unnecessary.
In this paper, the results section contains information about psychotropic drugs, but the contents of figures 1 to 5 should be described in the results section instead.
Discussion
The purpose of the discussion section is to reinforce the findings from this case and develop the discussion. The subject of this paper is the treatment of burns in EDS patients, particularly the treatment progress of skin grafts and LASCA. Therefore, the discussion section should delve deeper into these treatments.
The importance of psychological support for EDS patients is not denied, but this is also a common issue for non-EDS patients and is not a treatment specific to EDS. Therefore, it is not something that should be addressed at the beginning of the discussion section.
In addition, the authors do not express their views on the quote "Surgical complications in EDS generally fall into three categories" on page 13, line 230. Consideration should be given to how this quote influenced the treatment of this case.
Conclusion
In the conclusion, the authors emphasize "the importance of team medical care in the treatment of burn patients," but this is not closely related to the title of the paper. The conclusion should focus on the main theme of this paper: the usefulness of LASCA.

Author Response
Coments 1: "This is a valuable case report that reports on the treatment progress of a patient with Ehlers-Danlos syndrome (EDS) who sustained burns. However, this paper has the following problems. First, it is not formatted as a case report. Second, the viewpoint of the paper is not specific to EDS, but is more general.
In case reports, it is necessary to avoid general descriptions as much as possible and to proceed with discussion based on the information obtained from the case. It is also necessary to aim for concise and clear descriptions.
Abstract
The following sentences are not necessary in the Abstract or should be expressed concisely.
"joint hypermobility and dislocation, skin hyperelasticity and fragility with abnormal scar formation, and other features such as aortic dissection, blood vessel fragility, organ rupture, chronic pain and often"
"Additionally, the patient suffered from a chronic, iatrogenic wound and numerous laparotomies with partial resection of the sigmoid colon and a colostomy."
This is not directly related to burns or treatment, so the need to include it in the Abstract should be reconsidered.
"She was in a poor mental state. Since 2020, she has been taking antidepressant and anti-anxiety medication prescribed by a psychiatrist."
While the assessment of mental status and the presence or absence of treatment are clinically important, they are not specific to EDS, so their description should be kept to a minimum.
In addition, since "ADM" on line 27 is a first appearance, it should be spelled out in full as "acellular dermal matrix."
Conclusion
The following statements are inappropriate or unnecessary.
"Treating a patient with EDS is not the easiest thing to do."
This is self-evident and not necessary for the conclusion.
"During the patient's hospitalization, many adjustments were made to the treatment, one of the most important being the temporary discontinuation of medications prescribed by a psychiatrist."
This is not specific to EDS, so it is inappropriate for the conclusion.
There have been few reports of ADM or skin grafting in EDS patients, and the usefulness of specific treatments in the treatment of burns in EDS should be described based on the treatment course of this case.
- Introduction
The purpose of the introduction is to briefly explain the underlying disease that is the background of the case report and to clarify the uniqueness and specificity of this case. However, the introduction of this paper is limited to a detailed explanation of EDS (contents of Tables 1 and 2), and the introduction of LASCA (Laser-Assisted Skin Cell Application) in EDS, as stated in the title of the paper, is missing.
In addition, since this case report does not describe the subtype of EDS or the results of genetic testing, Tables 1 and 2 are considered unnecessary. Instead, the characteristics of LASCA and the circumstances that led to the reporting of this case should be clearly stated as an introduction.
2.1 Case Presentation
The description in 2.1 is redundant and difficult to read. One reason for this is the repeated use of the same words. In particular, the frequent use of "paraffin gauze and antiseptic" should be considered as to whether it is necessary to mention it repeatedly.
In addition, on page 7, line 109, it is stated that "a skin graft was obtained using a dermatome and transferred to the Tissue Bank located at the Burns Treatment Center for cell culture." However, there are few reports of allogeneic skin grafts for EDS patients. As stated in reference 13,
the donor site, total grafted area, and healing time should be detailed.
In addition, Figures 1 to 5 only contain descriptions of the time and location (department) of the images, and there is insufficient explanation of the findings. If there are findings specific to EDS patients, they should be clearly stated. In addition, these figures are not related to psychiatric prescriptions and are unnecessary.
In this paper, the results section contains information about psychotropic drugs, but the contents of figures 1 to 5 should be described in the results section instead.
Response 1: "Unfortunately, the patient never underwent genetic testing to determine the specific type of Ehlers-Danlos syndrome (EDS), which is why this information could not be included in the publication. However, given the clinical significance of the issue, it is crucial for physicians seeking reliable information in scientific publications—especially those who have never encountered a burn patient with EDS—to understand the broad scope of the problem.
From the very beginning of hospitalization, the medical staff paid particular attention to the patient's condition. As a major burn treatment center, we had never before encountered a patient with EDS who also had such extensive burn wounds. We recognized that this was a highly valuable case that could assist other physicians in finding reliable scientific information to develop the most optimal treatment protocol.
A detailed explanation of the significance of individual colors in the microperfusion imaging is provided in the materials and methods section, subsection 2.2 Imaging using Laser Speckle Contrast Analysis technique (lines 188-211). Additionally, the article includes further clarifications demonstrating the rationale for using laser microperfusion measurements throughout the hospitalization process (introduction, lines 81-87; materials and methods, lines 238-243; discussion, lines 283-289).
The article has also been updated with information about the donor site from which the skin sample was taken for cell culture (lines 138-148). This was one of the few areas on the patient’s body that remained unburned. Furthermore, we have included details regarding the transplantation of cultured cells during the surgical procedure (results section, lines 234-241).
Due to the exceptionally long hospitalization period (the patient was discharged after 182 days), every available method was employed in an attempt to aid the patient. This was a desperate effort to provide assistance. The materials and methods section describes the moment the patient was transferred from the surgical ward to the intensive care unit due to her deteriorating condition.
Thanks to systematic photographic documentation and LASCA measurements, an improvement in wound condition was observed. Previously used techniques (PRP, MEEK, autologous grafts) finally began to yield positive results—wounds started healing, and the patient’s condition improved day by day. As a result of this progress, the patient was transferred back to the surgical ward, where the previously applied treatments, including antidepressant medications, were reinstated. The medical staff was thus able to correlate the patient’s use of antidepressants with wound healing progression, as illustrated in Figure 1, Figure 4, and Figure 5."
Coments 2:
"Discussion
The purpose of the discussion section is to reinforce the findings from this case and develop the discussion. The subject of this paper is the treatment of burns in EDS patients, particularly the treatment progress of skin grafts and LASCA. Therefore, the discussion section should delve deeper into these treatments.
The importance of psychological support for EDS patients is not denied, but this is also a common issue for non-EDS patients and is not a treatment specific to EDS. Therefore, it is not something that should be addressed at the beginning of the discussion section.
In addition, the authors do not express their views on the quote "Surgical complications in EDS generally fall into three categories" on page 13, line 230. Consideration should be given to how this quote influenced the treatment of this case.
Conclusion
In the conclusion, the authors emphasize "the importance of team medical care in the treatment of burn patients," but this is not closely related to the title of the paper. The conclusion should focus on the main theme of this paper: the usefulness of LASCA."
Response 2: "We have made an effort to expand on the topics that were not described in sufficient depth and to improve the content of the Discussion, Results, and Conclusions sections. All changes have been highlighted in yellow."
Reviewer 3 Report
Comments and Suggestions for Authors
Ms. Ref. No.: medicina-3463054
Title: Successful treatment of patient with Ehlers Danlos Syndrome 2 (EDS) after extensive burn injury. Treatment carried out under 3 the supervision of speckle laser (LASCA): case report
Karolina Ziółkowska , Anna Słaboń , Justyna Glik, et al.
Overview and general recommendation:
Extensive burn injuries pose an interdisciplinary and interprofessional challenge to the treatment team, the victim’s family members, and the patient themselves. Even without significant pre-existing conditions, the treatment can be complex and demanding.
Karolina Ziólkowka and colleagues present a highly complex case in this report. They describe a patient with 60% total body surface area burned, who sustained her injuries in June 2021 and has since been under prolonged and complicated recovery at their clinic (since June 2022). Notably, the patient had a pre-existing diagnosis of Ehlers-Danlos syndrome, a connective tissue disorder that can lead to delayed wound healing.
In their article, the authors aim to address the various aspects of the complex treatment of a severe burn injury within their clinical setting. Interesting elements include the use of acellular dermal matrix (ADM), platelet-rich plasma (PRP) injection, and MEEK grafting, in addition to the utilization of Laser Speckle Contrast Analysis (LASCA). Furthermore, the authors attempt to comprehensively cover aspects of intensive care therapy, such as parenteral nutrition, as well as psychiatric support during the treatment process.
However, I have noticed that some sections of the paper provide excessive detail, while other key points are either insufficiently addressed or entirely missing. As a result, several questions have arisen that make it difficult for me to recommend this paper for publication in its current form. I believe a major revision is necessary, and I will outline my concerns in more detail below. I would appreciate it if the authors could specifically address these points in their response.
1.1 Major Comments
1.1.1 The abstract includes information that is not addressed or explained in the main body of the paper (e.g., multiple laparotomies, iatrogenic wounds, accident details). The purpose of an abstract is to provide the reader with an overview of the expected and discussed points, so it should not contain elements that are not covered later in the paper. Please either shorten the abstract or include these points in more detail within the body of the paper.
1.1.2 During my review of the paper, I was particularly struck by its complexity, as various aspects are addressed:
a. The introduction places a strong emphasis on Ehlers-Danlos syndrome and includes a large table (Table I.) from another source. A more concise formatting would improve clarity. It might be worth reconsidering whether all the information in this table is necessary for the reader's understanding of this case report.
b. The case presentation covers a wide range of topics, including various wound therapies, surgical concepts, the implementation of parenteral nutrition in the intensive care unit, and the discontinuation of psychotropic medication. Additionally, it contains striking surface images that are well-placed and appropriately labeled.
c. In the results section, the focus shifts to pain management, which was not mentioned earlier, as well as the patient’s overall psychological situation.
d. The first sentence of the discussion is not sufficiently explained earlier in the report: Why did various antidepressants and anxiolytic medications have the greatest impact on the wound healing process?
I can infer what the authors aim to convey with this paper. However, the sheer volume of information presented makes it difficult to grasp and the structure somewhat confusing, which is likely due to the complexity of the case. In my opinion, it would be beneficial to focus on two or three key aspects of the case and explain them in detail and with clarity. To assist the authors, perhaps the following questions could help guide their revision:
- How did the knowledge of the underlying condition influence the choice of surgical treatments? What led to the decision to use PRP or apply the MEEK technique?
- How does Speckle Laser (as described in the text) work, and how were the image analyses integrated into the therapy?
- What is the impact of antidepressant medications on wound healing, particularly in patients with Ehlers-Danlos syndrome, as supported by relevant sources?
1.2 Minor Comments
1.2.1 Page 2, Lines 53-55: Could you clarify what you mean by this? Specifically, how was attention given to the important role of fibroblasts and other relevant factors in this approach? This point seems to require further explanation to ensure it is understood in the context of your treatment plan or methodology.
1.2.2 Page 8, Lines 142-146: What new insights are gained by pausing and then restarting the psychotropic medication, as described in the figure caption? Could you provide further clarification on how this process contributes to the understanding or treatment of the case?
1.2.3 Pages 10 + 11, Figures 4 & 5: Please explain once to the reader what the colors in the images represent. Not every reader will be familiar with the procedure and may not be able to interpret the images correctly.
Author Response
Comments 1: "Overview and general recommendation:
Extensive burn injuries pose an interdisciplinary and interprofessional challenge to the treatment team, the victim’s family members, and the patient themselves. Even without significant pre-existing conditions, the treatment can be complex and demanding.
Karolina Ziólkowka and colleagues present a highly complex case in this report. They describe a patient with 60% total body surface area burned, who sustained her injuries in June 2021 and has since been under prolonged and complicated recovery at their clinic (since June 2022). Notably, the patient had a pre-existing diagnosis of Ehlers-Danlos syndrome, a connective tissue disorder that can lead to delayed wound healing.
In their article, the authors aim to address the various aspects of the complex treatment of a severe burn injury within their clinical setting. Interesting elements include the use of acellular dermal matrix (ADM), platelet-rich plasma (PRP) injection, and MEEK grafting, in addition to the utilization of Laser Speckle Contrast Analysis (LASCA). Furthermore, the authors attempt to comprehensively cover aspects of intensive care therapy, such as parenteral nutrition, as well as psychiatric support during the treatment process.
However, I have noticed that some sections of the paper provide excessive detail, while other key points are either insufficiently addressed or entirely missing. As a result, several questions have arisen that make it difficult for me to recommend this paper for publication in its current form. I believe a major revision is necessary, and I will outline my concerns in more detail below. I would appreciate it if the authors could specifically address these points in their response.
Response 1:
"Dear Reviewer,
We sincerely thank you for taking the time to review our manuscript and for your valuable comments. We appreciate that you have acknowledged the complexity of the treatment process and our efforts to utilize all available methods in our center to provide comprehensive care for the patient and facilitate wound closure.
Comments 2: "1.1 Major Comments
1.1.1 The abstract includes information that is not addressed or explained in the main body of the paper (e.g., multiple laparotomies, iatrogenic wounds, accident details). The purpose of an abstract is to provide the reader with an overview of the expected and discussed points, so it should not contain elements that are not covered later in the paper. Please either shorten the abstract or include these points in more detail within the body of the paper."
Response 2:
"1.1.1 In the discussion section (lines 263-269), we referred to the procedures initially performed at the Burn Treatment Center in Gryfice. The exact details of the accident are unknown, as the patient was in shock immediately afterward and could only recall the events leading up to the incident."
Comments 3: "1.1.2 During my review of the paper, I was particularly struck by its complexity, as various aspects are addressed:
- The introduction places a strong emphasis on Ehlers-Danlos syndrome and includes a large table (Table I.) from another source. A more concise formatting would improve clarity. It might be worth reconsidering whether all the information in this table is necessary for the reader's understanding of this case report."
Response 3: "We greatly appreciate your valuable suggestion. However, given the clinical significance of this issue, we believe it is important for the reader to receive a comprehensive understanding of the discussed condition. We consider it essential to present the full spectrum of EDS subtypes, as this may be beneficial for clinicians who consult our article for guidance on treating EDS patients—even in cases where the specific subtype remains unidentified due to the absence of genetic testing."
Comments 4: "The case presentation covers a wide range of topics, including various wound therapies, surgical concepts, the implementation of parenteral nutrition in the intensive care unit, and the discontinuation of psychotropic medication. Additionally, it contains striking surface images that are well-placed and appropriately labeled."
Response 4: "Thank you for acknowledging the quality of our figures. Throughout the hospitalization process, we made every effort to create highly detailed medical documentation. We were fully aware of how rarely surgeons encounter burn wounds in EDS patients. Our goal was to provide an accurate and thorough presentation of our patient’s hospitalization, which ultimately resulted in a successful outcome."
Comments 5: "In the results section, the focus shifts to pain management, which was not mentioned earlier, as well as the patient’s overall psychological situation."
Response 5: "The topic of pain management has been expanded in the abstract (lines 21-23), as well as in the materials and methods section (lines 133-134, 244-250). The issue of antidepressant treatment has been addressed in the materials and methods section (lines 131-132, 250-255)."
Comments 6: "The first sentence of the discussion is not sufficiently explained earlier in the report: Why did various antidepressants and anxiolytic medications have the greatest impact on the wound healing process?
I can infer what the authors aim to convey with this paper. However, the sheer volume of information presented makes it difficult to grasp and the structure somewhat confusing, which is likely due to the complexity of the case. In my opinion, it would be beneficial to focus on two or three key aspects of the case and explain them in detail and with clarity. To assist the authors, perhaps the following questions could help guide their revision:
- How did the knowledge of the underlying condition influence the choice of surgical treatments? What led to the decision to use PRP or apply the MEEK technique?
- How does Speckle Laser (as described in the text) work, and how were the image analyses integrated into the therapy?
- What is the impact of antidepressant medications on wound healing, particularly in patients with Ehlers-Danlos syndrome, as supported by relevant sources?"
Response 6: "The manuscript presents various therapeutic methods aimed at closing the burn wound. The MEEK technique is typically used in patients with limited donor site availability, which was the case for our patient. Due to the exceptionally long hospitalization period (the patient was discharged after 182 days), every available method was employed to aid the patient. This was a desperate attempt to provide assistance.
The materials and methods section describes the transfer of the patient from the surgical ward to the intensive care unit due to her deteriorating condition. Thanks to systematic photographic documentation and LASCA measurements, an improvement in wound condition was observed. Previously used techniques (PRP, MEEK, autologous grafts) finally began to yield positive results—wounds started healing, and the patient's condition improved day by day.
As a result of this progress, the patient was transferred back to the surgical ward, where the previously applied treatments, including antidepressants, were reinstated. The medical staff was thus able to correlate the patient's use of antidepressant medication with treatment progression, as illustrated in Figure 1, Figure 4, and Figure 5.
The application and use of the speckle laser technique in monitoring treatment progress has been discussed in multiple sections of the text, including figure descriptions (Figure 4 and Figure 5), the materials and methods section (subsection 2.2, lines 241-243), and the discussion (lines 283-289 and 327-330)."
Comment 7: " 1.2 Minor Comments
1.2.1 Page 2, Lines 53-55: Could you clarify what you mean by this? Specifically, how was attention given to the important role of fibroblasts and other relevant factors in this approach? This point seems to require further explanation to ensure it is understood in the context of your treatment plan or methodology."
Response 7: "The treatment of deep burns has been extensively elaborated in the introduction (lines 57-80). Additionally, we have included a description of treatment using ADM grafts as well as advanced therapy medicinal products under the hospital exemption (ATMP-HE)."
Comment 8: "1.2.2 Page 8, Lines 142-146: What new insights are gained by pausing and then restarting the psychotropic medication, as described in the figure caption? Could you provide further clarification on how this process contributes to the understanding or treatment of the case?"
Response 8: "1.2.2 Due to the exceptionally long hospitalization period (the patient was discharged after 182 days), every available method was employed in an attempt to aid the patient. This was a desperate effort to provide assistance. The materials and methods section describes the moment when the patient was transferred from the surgical ward to the intensive care unit due to her deteriorating condition.
Thanks to systematic photographic documentation and LASCA measurements, an improvement in wound condition was observed. Previously used techniques (PRP, MEEK, autologous grafts) finally began to yield positive results—wounds started healing, and the patient’s condition improved day by day. As a result of this progress, the patient was transferred back to the surgical ward, where the previously applied treatments, including antidepressants, were reinstated. The medical staff was thus able to correlate the patient’s use of antidepressant medication with treatment progression, as illustrated in Figure 1, Figure 4, and Figure 5. "
Comments 9: "1.2.3 Pages 10 + 11, Figures 4 & 5: Please explain once to the reader what the colors in the images represent. Not every reader will be familiar with the procedure and may not be able to interpret the images correctly."
Response 9: "1.2.3 A detailed explanation of the significance of individual images is provided in the materials and methods section, subsection 2.2 Imaging using Laser Speckle Contrast Analysis technique (lines 189-195)."
We sincerely appreciate the time you have dedicated to reviewing our manuscript, as well as your valuable comments and positive evaluation of our work. Best regards, Anna Słaboń.
Round 2
Reviewer 2 Report
Comments and Suggestions for Authors
Reviewer's comments
I have read the revised version of the paper.
Although some parts have been appropriately revised, the following revisions are necessary.
Points that require reconsideration
① Authors are obligated to respond to each and every comment made by the reviewers. However, it is not necessary to follow the reviewer's instructions, and it is also appropriate to refute them with evidence. In light of this, the results of the first review, which did not receive a response, are also reprinted below.
② The authors emphasize in the abstract and discussion that the use of psychotropic drugs has a negative effect on the burn healing process in EDS patients. However, in this case, it is stated in the first draft (lines 119-120) that these psychotropic drugs were already being taken before the patient was admitted to the ICU. Furthermore, although Reference 18 is cited as evidence, the cause of vasculitis in Reference 18 is sulfonamide, not psychotropic drugs. In addition, in Reference 18, symptoms appeared promptly after the start of sulfonamide medication, which does not match this case.
Regarding this point, the authors responded, "The medical staff was thus able to correlate the patient’s use of antidepressants with wound healing progression, as illustrated in Figure 1, Figure 4, and Figure 5."
However, to conclude that psychotropic drugs adversely affected the burn healing process in EDS patients, objective medical evidence (findings from other papers) is required. It is inappropriate to emphasize this point in the paper without sufficient evidence.
③ The title of this paper focuses on the running title, "the supervision of laser speckle contrast analysis." However, the conclusion in the abstract does not mention this point. The emphasis is on the effects of psychotropic drugs, which does not match the title of this paper.
④ The authors provide details of EDS subtypes and the genetic changes associated with the subtypes in Tables 1 and 2. However, considering that this is a case report, the subtype of this case is unknown, and the involvement of the subtype in the clinical course and treatment of this case is not mentioned in the paper, wouldn't a citation of the literature be sufficient? Similar statements can be found in various places in the text.
⑤ The content of lines 263-269 of the Discussion section is the clinical course and does not qualify as a consideration.
⑥ The content of lines 270-271 is self-evident and does not need to be included in the Discussion section of this case. Emotional expressions should also be avoided.
⑦ In lines 272-289, the reviewer believes that the most important point in the discussion section of this paper is the consideration of the usefulness of these various treatments and the evaluations of LASCA in treating burns in EDS patients. This point needs to be reconsidered.
⑧ Although abbreviations such as acellular dermal matrix (ADM) and Ehlers-Danlos syndrome (EDS) are presented in the text, the full spelling is repeatedly used in places, and abbreviations are not used in places where they should be used. Please correct as appropriate.
⑨ It is not necessary to use the abbreviation (ATMPs) for words that appear for the first time in the Conclusion.

Author Response
Reviewer's comments I have read the revised version of the paper. Although some parts have been appropriately revised, the following revisions are necessary. Points that require reconsideration
① Authors are obligated to respond to each and every comment made by the reviewers. However, it is not necessary to follow the reviewer's instructions, and it is also appropriate to refute them with evidence. In light of this, the results of the first review, which did not receive a response, are also reprinted below.
Dear Reviewer,
Thank you for highlighting this important issue, as it has allowed us to elevate our work to a higher level. Moreover, your comments made us realize that we are likely the first to make such observations. We have made every effort to address your review as accurately and thoroughly as possible. We sincerely hope that this time our revisions and responses will be satisfying.
② The authors emphasize in the abstract and discussion that the use of psychotropic drugs has a negative effect on the burn healing process in EDS patients. However, in this case, it is stated in the first draft (lines 119-120) that these psychotropic drugs were already being taken before the patient was admitted to the ICU. Furthermore, although Reference 18 is cited as evidence, the cause of vasculitis in Reference 18 is sulfonamide, not psychotropic drugs. In addition, in Reference 18, symptoms appeared promptly after the start of sulfonamide medication, which does not match this case. Regarding this point, the authors responded, "The medical staff was thus able to correlate the patient’s use of antidepressants with wound healing progression, as illustrated in Figure 1, Figure 4, and Figure 5." However, to conclude that psychotropic drugs adversely affected the burn healing process in EDS patients, objective medical evidence (findings from other papers) is required. It is inappropriate to emphasize this point in the paper without sufficient evidence.
Thank you for your valuable comment. We agree that objective medical evidence is necessary to assess the impact of psychotropic drugs on the wound healing process in EDS patients. In our study, we relied on clinical observations suggesting a correlation between the use of antidepressants and the progression of wound healing. However, we recognize that for the full credibility of this conclusion, more robust evidence is required. Unfortunately, there is a deficit of scientific literature addressing the treatment of wounds in EDS patients taking psychotropic medications. However, our observations were unequivocal. In that specific aproach we associated the observed negative impact of the drugs on wound condition with another disease syndrome, in which we have extensive experience—Lyell's syndrome (toxic epidermal necrolysis TEN). There are several reports in the literature regarding the impact of drugs on the occurrence of toxic epidermal necrolysis, also known as Lyell’s syndrome. In response to your comment, we included additional references in our work that support our thesis. As you pointed out, our patient was already taking psychiatric medications prior to admission to the intensive care unit. We believe this is an important aspect to mention, as in healthy individuals with no prior issues with wound healing, burn wounds should gradually begin to close, especially after the use of autologous grafts, MEEK method grafts, cultured keratinocyte grafts from the tissue bank, and the application of ADM, as well as following the attending physician's recommendations. Despite all efforts made by the hospital staff, the wounds did not heal at all or healed only to a minimal extent. It was only after discontinuing the psychotropic medications, associated with the patient’s transfer to the intensive care unit, that the grafts began to take and the wounds started to decrease in size.
③ The title of this paper focuses on the running title, "the supervision of laser speckle contrast analysis." However, the conclusion in the abstract does not mention this point. The emphasis is on the effects of psychotropic drugs, which does not match the title of this paper.
Thank you for pointing out this aspect. We have modified the title of our paper. We have removed the second sentence that mentioned the use of the speckle laser during the patient's hospitalization. Additionally, the conclusions in the abstract have also been revised.
④ The authors provide details of EDS subtypes and the genetic changes associated with the subtypes in Tables 1 and 2. However, considering that this is a case report, the subtype of this case is unknown, and the involvement of the subtype in the clinical course and treatment of this case is not mentioned in the paper, wouldn't a citation of the literature be sufficient? Similar statements can be found in various places in the text.
Thank you for that comment however the tables you have mentioned are significant for the theoretic backround of the work. The explanation is provided in lines 47 to 52. The tables included in the introduction are a crucial element as they illustrate the scale of the problem. In response to the reviewer’s request, the tables that were previously part of the introduction have been moved to the supplementary material.
The presented case report is submitted by our team from a specialized burn treatment center. We are a monospecialist hospital, and genetic testing for the confirmation or exclusion of Ehlers-Danlos syndrome (EDS) is not routinely performed in our facility. Additionally, we do not classify EDS based on genetic studies. The treatment of burn wounds resulting from thermal injury follows a standardized protocol.
⑤ The content of lines 263-269 of the Discussion section is the clinical course and does not qualify as a consideration.
Lines 263 to 269 have been removed.
⑥ The content of lines 270-271 is self-evident and does not need to be included in the Discussion section of this case. Emotional expressions should also be avoided.
The content previously included in lines 270 to 271 has been modified and highlighted in yellow.
⑦ In lines 272-289, the reviewer believes that the most important point in the discussion section of this paper is the consideration of the usefulness of these various treatments and the evaluations of LASCA in treating burns in EDS patients. This point needs to be reconsidered.
After reviewing your comments, we have made a change to the title of the paper. We have removed the reference to monitoring wound healing progress using speckle laser technology. Our intention was never to mislead the reader. In the 'Discussion' section, we aimed to highlight the wide range of methods used to facilitate wound closure in a patient with extensive wounds covering nearly the entire body. All therapeutic methods we employed are routinely used in our hospital to ensure the highest possible quality of life for patients after discharge.
⑧ Although abbreviations such as acellular dermal matrix (ADM) and Ehlers-Danlos syndrome (EDS) are presented in the text, the full spelling is repeatedly used in places, and abbreviations are not used in places where they should be used. Please correct as appropriate.
Thank you for your valuable comment. We have carefully reviewed the text and corrected the inconsistent use of abbreviations. The full terms are now used only at their first mention, followed by the appropriate abbreviations throughout the rest of the manuscript. We appreciate your feedback, which has helped improve the clarity and consistency of our work.
⑨ It is not necessary to use the abbreviation (ATMPs) for words that appear for the first time in the Conclusion
Thank you for your comment. However, we believe that the abbreviations should remain in the conclusion so that the reader, who may not be familiar with the techniques we use, can better navigate the content of our work.
Reviewer 3 Report
Comments and Suggestions for Authors Thank you for considering my comments. The complexity of the case could now be worked out even better and the “take home messages” are now addressed in a clearly comprehensible manner. A great article! Thank you for your trust and your great work!Author Response
Thank you for considering my comments. The complexity of the case could now be worked out even better and the “take home messages” are now addressed in a clearly comprehensible manner. A great article! Thank you for your trust and your great work!
Thank you very much for your kind words and valuable feedback. We truly appreciate your thoughtful review and the time you dedicated to improving our manuscript. Your comments were extremely helpful in refining our work, and we are delighted that the key messages are now conveyed more clearly. It has been a pleasure to have your insights, and we are grateful for your support and encouragement. Thank you once again!